# SILK: Smooth InterpoLation frameworK for motion in-betweening
# A Simplified Computational Approach

## Abstract

*Motion in-betweening is a crucial tool for animators, enabling intricate control over pose-level details in each keyframe. Recent machine learning solutions for motion in-betweening rely on complex models, incorporating skeleton-aware architectures or requiring multiple modules and training steps. In this work, we introduce a simple yet effective Transformer-based framework, employing a single Transformer encoder to synthesize realistic motions in motion in-betweening tasks. We find that data modeling choices play a significant role in improving in-betweening performance. Among others, we show that increasing data volume can yield equivalent or improved motion transitions, that the choice of pose representation is vital for achieving high-quality results, and that incorporating velocity input features enhances animation performance. These findings challenge the assumption that model complexity is the primary determinant of animation quality and provide insights into a more data-centric approach to motion interpolation. Additional videos and supplementary material are available at https://silk-paper.github.io.*

## 1. Introduction

In this work, we focus on developing a deep learning-based approach for solving animation in-betweening. Animation in-betweening is the process of generating intermediate frames to smoothly transition between artist-specified keyframes, creating the illusion of natural movement. Keyframing and in-betweening remain the primary way to create believable animations while allowing animators to control pose-level details of each keyframe. Existing in-betweening software often fails to generate realistic transitions for large keyframe gaps, necessitating animators to manually insert additional keyframes. With the rise of deep learning, several learning-based solutions have been proposed to solve the long horizon in-betweening problem, e.g., [9, 15, 16, 42].

The most successful machine learning models to-date are Transformer-based approaches that treat in-betweening as a sequence-to-sequence problem [22, 24, 26]. In order to create a sequence-to-sequence setup, the missing input frames are pre-filled either with linearly interpolated values [2, 7, 36–38] or the predictions of a separate Transformer model [26] while learnable position embeddings encode the positions of missing frames. Training a single Transformer encoder in this setup has so far only achieved inferior results [7, 16] compared to more complex models with several encoders or training stages [22, 24, 26].

In this work, we show that data modeling choices and the quantity of data play a more significant role than architecture design. Contrary to previous findings, we demonstrate that a single Transformer encoder can solve the in-betweening task as well as complex models. We propose a simplified approach to motion in-betweening that achieves comparable or superior results to state-of-the-art methods while significantly reducing architectural complexity. Our key findings are:

- We demonstrate that a single transformer architecture performs as well as or better than complex models in in-betweening tasks.
- We employ only relative position embeddings without any additional embedding types (such as keyframe position embeddings), demonstrating that a single, simple embedding strategy is sufficient.
- Contrary to the vast majority of related work, we find that interpolation in the input sequence leads to inferior results compared to simple zero filling.
- We demonstrate that creating a more extensive dataset by sampling data points with a smaller offset than used by the related work significantly increases performance for simpler models.
- We show that operating directly in trajectory space can have several advantages compared to working in local-to-parent space.
- We find that including velocity features in the model input boosts the model's capabilities.

## 2. Related Work

Motion in-betweening refers to the constrained motion synthesis problem in which the final character pose is known in advance, and the generation time window is also specified. Methods to automatically fill the gaps between hand-drawn keyframes have a long history. For example, [18] introduced an automatic stroke-based system to interpolate between hand-drawn keyframes.

Time series models such as recurrent neural networks (RNNs) have been the main approach for several years. [40] introduced an RNN-based system to autocomplete keyframes for a simple 2D character. [8] introduced Recurrent Transition Networks (RTN) to autocomplete more complex human motions with a fixed in-betweening length of 60 frames. The key idea is to use two separate encoders, a target encoder and an offset encoder, to encode parts of the future context. RTNs have been extended by [42] to allow for variable in-betweening lengths ranging from one to thousands of frames. Concurrently, [9] enhanced RTN by adding a time-to-arrival embedding to support varying transition lengths, as well as using scheduled target noise to allow variations and improve robustness in generated transitions. Both [9] and [42] leveraged adversarial training to further improve the quality of the generated motion. The motion manifold model [32] conditions motion transitions on control variables, allowing for transition disambiguation while simultaneously ensuring high-quality motion. As a non-recurrent model, [15] proposed an end-to-end trainable convolutional autoencoder for filling in missing frames. Our proposed model, in contrast to previous RNN-based or convolution-based methods, employs Transformers to handle various forms of missing frames in parallel through a single-shot prediction.

Several generative models have been designed that also support in-betweening, e.g., based on Variational Autoencoders [10] and diffusion models [33]. [17] propose a framework that allows for arbitrary sparse keyframe placement by leveraging diffusion models. It achieves diverse and coherent motion synthesis, albeit at the cost of increased inference time and the need for repeated guidance mechanisms to enforce constraints.

Guided Motion Diffusion (GMD) [14] incorporates spatial constraints into motion synthesis, enabling control over global motion trajectories, sparse keyframe placements, and obstacle avoidance. However, GMD relies on imputation and inpainting techniques, which struggle to consistently maintain spatial constraints over longer sequences. Similarly, Factorized Motion Diffusion (FMD) [30] introduces a model that factorizes motion into character-agnostic Bezier curves followed by an inverse kinematics module. While this allows for precise control of sparse constraints, FMD requires an additional step of pose reconstruction, potentially introducing errors and limiting generalization across different character topologies. As we are focusing on deterministic motion in-betweening, we will not directly compare our model to these methods.

### 2.1. Transformer-based Motion In-betweening

A number of Transformer-based approaches have been suggested that treat in-betweening as a sequence-to-sequence problem. These solutions present different versions of a set of design choices such as how to represent the input and output data, how to encode frame embeddings, whether and how to pre-fill the missing frames, whether to use additional constraining keyframes during training, the model architecture and the loss function.

As one of the first Transformer-based in-betweening models, [16] treats motion in-betweening as a masked motion modeling problem, similar to BERT [6] in computational linguistics. [16] aim at a model that can be conditioned both on keyframes and semantic information, such as motion types, to produce the intermediate frames. As a sequence-to-sequence Transformer model operates on both the present keyframes and the missing frames, they propose to fill the missing frames with a linear interpolation between the start and end keyframes. Many of the works following [16] that we describe below employ the same linear interpolation strategy for the input values [1, 2, 7, 36, 38]. Extending [16], [7] propose learnable position encoding for keyframes and learn additional embeddings that encode whether a frame is a keyframe, missing or an ignored frame. The model architecture includes a convolutional layer before and after the transformer to encode the temporal structure of the data, and they output joint positions and rotations directly. We simplify this model by removing the convolutional layers and the additional embeddings. To account for inter-joint relationships, [38] employs skeleton-aware [1] encoder and decoder networks to interpolate between dancing sequences. Similarly, [2] introduce skeleton awareness. Instead of the encoder and decoder, they propose a skeletal graph transformer operating on the joints of the skeleton. Another example is [37], who utilize spatio-temporal graph convolutional layers, skeleton pooling and unpooling layers to extract motion sequence features while employing a U-Net structure to integrate keyframe information. They test their model both with global positions and with quaternions as feature representation. Finally, [36] proposes a Spatial and Temporal Transformer for Motion In-Betweening (ST-TransMIB), a novel spatio-temporal transformer framework for motion in-betweening that is based on interpolated input sequences. Their approach introduces a spatial transformer to model per-frame joint interactions, with the aim to improve generalization across action types. Additionally, they design a multi-scale temporal transformer that captures both global motion trends and fine-grained local variations in order to mitigate over-smoothing. We show that using

zeros as input suffices for a simple transformer architecture to solve in-betweening without the need to model spatial correlations explicitly.

The Delta Interpolator introduced by [24] does not use linearly interpolated frames as input. Instead, the model learns to output the residual that is added to the interpolated frames to yield the final prediction. As the only input to the model are the keyframes, the model therefore needs to learn both linear interpolation and realistic human motion. Additionally, this complicates model design as it requires two separate encoders, one for existing keyframes and one for missing frames. [22] adopts the same output strategy as [24] based on residual predictions. They propose a Vision Transformers encoder - decoder model with a pose decomposition module.

Moving away from keyframe interpolation as input, [26] introduce two Transformer models, a Context and a Detail Transformer, to generate in-betweening frames at different fidelities. The Context Transformer is a encoder-only Transformer that is responsible for generating a rough outline of the transition motion based on the input context and the target frame. It thus replaces the need for additional interpolated inputs. The Detail Transformer is an encoder-only Transformer that refines the output of the Context Transformer. While this architecture performs well in experiments, it adds additional complexity to a rather simple learning problem and requires two separate training steps, one for each model.

In this work, we show that the added complexity in [22, 24, 26, 36–38] and [4] is unnecessary and that a single Transformer encoder suffices. Importantly, all previous methods use some form of global or local-to-parent position and rotation features to represent the pose, while we represent the pose in root space as discussed in Section 3.1.1.

## 3. Methodology

SILK is a non-autoregressive model that generates the desired number of in-between frames in a single shot given the context and target keyframes. We propose a minimalist and yet effective Transformer-based architecture for SILK.

### 3.1. Data Representation

Consider a dataset of $N$ animations denoted by $\{X_i\}_{i=1}^{N}$. Each animation $X_i$ consists of a sequence of $n_i$ frames. We represent $X_i$ as $X_i = [X_i^1, X_i^2, \ldots, X_i^{n_i}]$, where $X_i^j$ is a $d$-dimensional feature vector that corresponds to the $j$-th frame of the $i$-th animation. Here, $d$ is the number of features that represent each frame.

#### 3.1.1. Feature representation

Unlike many previous in-betweening works, e.g., [9, 26, 27], we diverge from the conventional use of the local-to-parent space joint features. We find that using root space

pose features, e.g., [11, 25, 39], as the network input results in a more stable training process and achieves higher quality synthesized motions (see Table 3). It is worth noting that while using root space pose features improves the overall motion quality, it comes at a cost of additional inverse kinematics passes at runtime to generate the required local-to-parent rotations. Although the IK computation can be parallelized across the entire sequence of generated frames, it can still be expensive for complex skeletons, especially if real-time user feedback is required.

To compute the root space pose features, we start by projecting the hip joint onto the ground, creating an artificial trajectory joint as the root node of the skeleton. This way, the root position can be represented as a 2-dimensional vector on the xz-plane and the root orientation can be represented as a single rotation about the y-axis. However, to avoid the discontinuity problem with Euler angles, we use the cosine-sine representation for the root orientation. To compute the rest of the joint features, we first find the 4x4 joint transformation matrices in global space using forward kinematics, and then we transform the global matrices into root space by multiplying with the inverse of the root transformation. Orientations are encoded using the 6D representation described in [41].

We distinguish between input and output features. In particular, output features do not include velocities. Each input frame is of dimension $d_{in} = (18J + 8)$, where $J$ corresponds to the number of joints in the skeleton. The input features are the concatenation of root position ($\mathbb{R}^2$), root orientation ($\mathbb{R}^2$), root linear velocity ($\mathbb{R}^2$), root angular velocity ($\mathbb{R}^2$), joint positions ($\mathbb{R}^{3J}$), joint orientations ($\mathbb{R}^{6J}$), joint linear velocities ($\mathbb{R}^{3J}$), and joint angular velocities ($\mathbb{R}^{6J}$). The output frame, on the other hand, has a dimension $d_{out} = (9J + 4)$, which is a concatenation of root position ($\mathbb{R}^2$), root orientation ($\mathbb{R}^2$), joint positions ($\mathbb{R}^{3J}$) and joint orientations ($\mathbb{R}^{6J}$).

### 3.2. SILK Architecture and Training Procedure

We assume that the model is given a $C$ context frames, the last context frame is considered the start keyframe, and one target keyframe. Providing $C$ context frames is common in the in-betweening literature as a means to guide the motion to follow a desired theme among all the possible in-between animations.

#### 3.2.1. Model input

The SILK architecture is shown in Figure 1. We provide the model with a sequence of frames that consists of $C$ context frames, $M$ missing frames filled with zeros, and a final target frame. $M$ is a variable parameter that is specified by the animator.

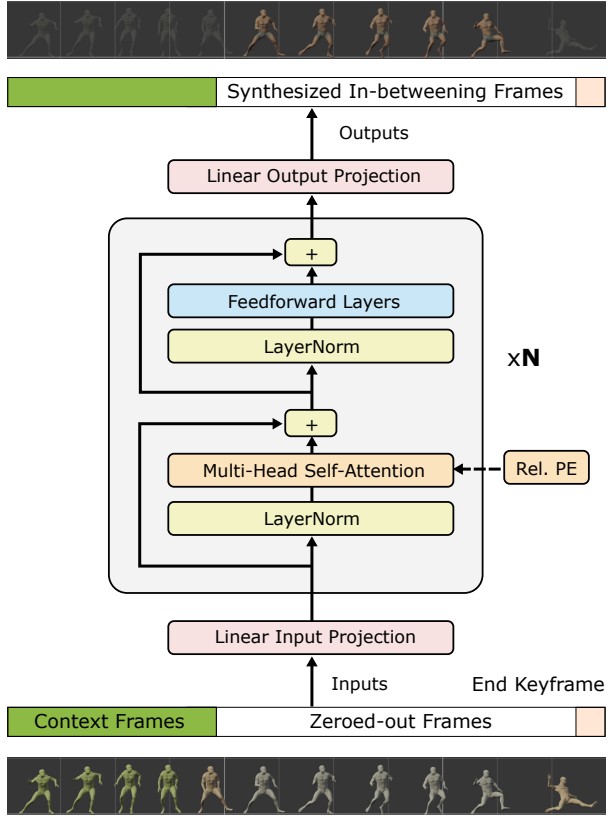

Figure 1. SILK's architecture. The missing frames are filled with zeroes and passed to a single Transformer encoder. No masking is performed such that all frames can attend to all other frames.

**Positional encoding** In order to equip the model with temporal information, we provide positional embeddings as an additional input. Following a similar approach to [26], we use learned relative positional encoding which represents positional information based on the relative distances between frames in the animation sequence. At inference, this allows the model to handle sequences that are longer than sequences seen during training.

### 3.2.2. Model architecture

We use a single Transformer encoder as shown in Figure 1. For details on Transformers, we refer the reader to [35]. The input to the Transformer encoder is the sequence of the frames prepared as described in Section 3.2.1. We linearly project each frame into the required input dimension prior to feeding the animation into the Transformer.

Note that we are not masking the in-between frames. Instead, all-zero frames are provided to the first layer of the Transformer. In every subsequent layer, the representation of each frame is allowed to attend to the representation of all the other frames, whether they are in-between frames or not. This allows for the predictions to remain faithful to the context frames and smooth with respect to neighbouring frames.

### 3.2.3. Model output

The output of the Transformer is mapped back to the original dimension $d_{out}$ using a linear mapping, forming the final animation.

### 3.2.4. Loss Function

We train the model using the L1 norm to measure the difference between the generated animations and the ground truth. Compared to the L2 norm, the L1 norm results in smoother learning curves throughout training and yields superior outcomes. Previous approaches often calculate separate losses for position, rotation, and foot sliding, requiring careful tuning of coefficients to balance these terms. In contrast, we simplify this by applying a single L1 loss across all features, eliminating the need for complex loss weighting while maintaining effective training performance.

## 4. Experiments

This section presents a comprehensive analysis of the SILK's performance across several evaluation criteria. It includes a comparison to state-of-the-art methods based on the evaluation protocol proposed in [9]. We also investigate the impact different training data sub-sampling strategies, pose feature representation, pre-filling of the missing frames and the use of velocity features.

### 4.1. Datasets

We train and evaluate our model on two different datasets. The majority of experiments focus on the Ubisoft LaFAN1 dataset [9]. We follow the same setup as [9] for processing the LaFAN1 dataset, which contains approximately 4.5 hours of locomotion, fighting, dancing, aiming, ground motions, falling, and recovery data. The key difference in our approach is the slice offset: we use an offset of 5 when segmenting the training animation data, compared to the offset of 20 used in previous works. This modification results in a training dataset four times larger than the original. We emphasize that the source training data remains identical—we are simply extracting more sub-animations, which we hypothesize will improve the model's performance by exposing it to a greater number of motion transitions and temporal relationships within the same underlying data.

### 4.2. Training and Hyperparameters

SILK is an encoder-style Transformer model consisting of 6 transformer layers and 8 attention heads per layer. Each layer contains the self-attention mechanism and a two-layer feedforward neural network. The model's internal representation size, $d_{model}$, is set to 1024 and the dimension of the feedforward layer, $d_{ff}$, is 4096. In the transformer architecture, we ensure that layer normalization takes place prior

Table 1. Comparing L2P, L2Q, and NPSS Metrics on the LaFAN1 Dataset Using Various Methods. We only include results that report comparable numbers for the SLERP and RMIB baselines as discrepancies indicate differences in metric computation (e.g. [5, 29]). Lower values indicate better performance. For easier comparison, we added the task that the respective model was designed to solve. Motion In-betweening (MIB) is the standard setting. Some models have been designed for real-time Motion In-betweening (RT-MIB), Motion In-betweening with semantic conditioning (S-MIB) or to solve a range of motion modeling tasks such as prediction. We call the latter group Motion In-betweening plus (MIB+).

| | Task | L2P ↓ | | | | L2Q ↓ | | | | NPSS ↓ | | | |
|---|---|---|---|---|---|---|---|---|---|---|---|---|---|
| Length (frames) | | 5 | 15 | 30 | 45 | 5 | 15 | 30 | 45 | 5 | 15 | 30 | 45 |
| SLERP | MIB | 0.37 | 1.38 | 2.49 | 3.45 | 0.22 | 0.62 | 0.97 | 1.25 | 0.0023 | 0.040 | 0.225 | 0.45 |
| RMIB [9] | MIB | 0.23 | 0.65 | 1.28 | 2.24 | 0.17 | 0.42 | 0.69 | 0.94 | 0.0020 | 0.026 | 0.133 | 0.33 |
| HHM [20] | MIB+ | - | - | - | - | 0.24 | 0.54 | 0.94 | 1.2 | - | - | - | - |
| UMC [7] | MIB+ | 0.23 | 0.56 | 1.06 | - | 0.18 | 0.37 | 0.61 | - | 0.0018 | 0.024 | 0.122 | - |
| CMIB [16] | S-MIB | - | - | 1.19 | - | - | - | 0.59 | - | - | - | - | - |
| RTC [32] | RT-MIB | 0.20 | 0.56 | 1.11 | - | - | - | - | - | 0.0055 | 0.070 | 0.346 | - |
| TWE [28] | MIB | 0.21 | 0.59 | 1.21 | - | 0.16 | 0.39 | 0.65 | - | 0.0019 | 0.026 | 0.136 | - |
| TST [26] | MIB | **0.10** | 0.39 | 0.89 | 1.68 | 0.10 | 0.28 | 0.54 | 0.87 | 0.0011 | 0.019 | 0.112 | 0.32 |
| Δ-I [24] | MIB | 0.13 | 0.47 | 1.00 | - | 0.11 | 0.32 | 0.57 | - | 0.0014 | 0.022 | 0.122 | - |
| CTL [23] | MIB | 0.30 | 0.71 | 1.26 | - | 0.21 | 0.40 | 0.63 | - | 0.0019 | 0.028 | 0.139 | - |
| DPS [27] | RT-MIB | - | 0.45 | 0.99 | 1.59 | - | 0.32 | 0.57 | 0.92 | - | 0.020 | 0.120 | **0.30** |
| DMT [4] | MIB | 0.12 | 0.49 | 1.04 | - | **0.09** | 0.31 | 0.52 | - | **0.0010** | 0.022 | 0.120 | - |
| UNI-M [22] | MIB+ | 0.14 | 0.46 | 0.97 | - | 0.12 | 0.32 | 0.57 | - | 0.0014 | 0.022 | 0.120 | - |
| DC [38] | MIB | - | - | 1.04 | - | - | - | 0.54 | - | - | - | 0.121 | - |
| STG [37] | MIB | - | 0.71 | - | - | - | 0.36 | - | - | - | 0.027 | - | - |
| ST-TMIB [36] | MIB | 0.13 | 0.40 | 0.89 | 1.62 | 0.11 | 0.29 | 0.55 | 0.84 | 0.0014 | 0.020 | 0.117 | 0.32 |
| SILK | MIB | 0.13 | **0.38** | **0.83** | **1.59** | 0.11 | **0.27** | **0.50** | **0.79** | 0.0012 | **0.018** | **0.105** | 0.30 |

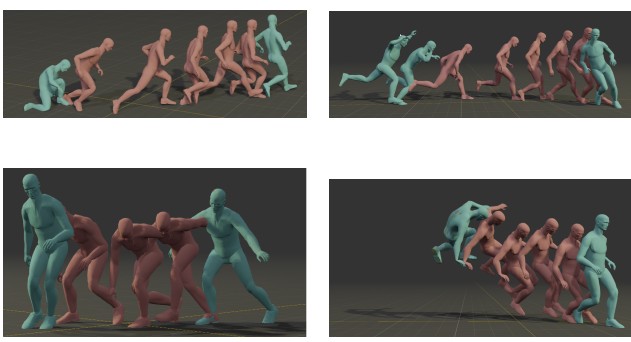

Figure 2. Four sample in-between animations generated by SILK based on the LAFAN1 dataset.

to attention and feedforward operations. The model is optimized using the AdamW optimizer with a noam learning rate scheduler same as [26]. We set the batch size to 64. We make use of the code provided by [26] to train benchmark models.

### 4.3. LaFAN1 Benchmark Evaluation

We evaluate our model on the LaFAN1 dataset using the same procedure as [9], namely, training on subjects 1 through 4 and testing on subject 5. The model is trained on samples with 10 context frames and one target keyframe. For training with variable length inbetweening frames, we uniformly sample the transition lengths between 5 and 30. At evaluation time, the transition length is fixed to 5, 15, 30 or 45. We follow previous work and compute three metrics: the normalized L2 norm of global positions (L2P), the L2 norm of global quaternions (L2Q), and normalized power spectrum similarity (NPSS) [9, 16, 26].

Table 1 shows the quantitative comparison between SILK and previous works. In summary, SILK demonstrates similar or superior performance for all metrics across nearly all different lengths. While the model was trained on sequences with up to 30 missing frames, its performance on the 45-frame condition demonstrates its ability to extrapolate effectively to an unseen number of frames.

For visual comparisons between TST and SILK, we refer the reader to the supplementary videos. In contrast to TST, it is important to note that we are not employing post-processing and removing foot-sliding. When comparing the predicted sequences by the two models, it is apparent that the predictions are often indistinguishable for shorter gaps (five and fifteen missing frames). As short gaps not allow for a lot of variability, models at a similar capability level

Table 2. Comparing L2P, L2Q, and NPSS Metrics on the LaFAN1 Dataset using different offsets when sampling training data.

| Model | Offset | 5 | 15 | 30 | 45 |
|-------|--------|-----|-----|-----|-----|
| | | \multicolumn{4}{c}{Length (frames)} | | | |
| | | \multicolumn{4}{c}{L2P ↓} | | | |
| SILK | 5 | 0.13 | 0.38 | 0.83 | 1.59 |
| SILK | 20 | 0.16 | 0.41 | 0.98 | 1.94 |
| TST [26] | 5 | 0.10 | 0.40 | 0.93 | 1.73 |
| TST [26] | 20 | 0.10 | 0.39 | 0.89 | 1.68 |
| | | \multicolumn{4}{c}{L2Q ↓} | | | |
| SILK | 5 | 0.11 | 0.27 | 0.50 | 0.79 |
| SILK | 20 | 0.12 | 0.28 | 0.56 | 0.93 |
| TST [26] | 5 | 0.10 | 0.29 | 0.56 | 0.86 |
| TST [26] | 20 | 0.10 | 0.28 | 0.54 | 0.87 |
| | | \multicolumn{4}{c}{NPSS ↓} | | | |
| SILK | 5 | 0.0012 | 0.018 | 0.105 | 0.30 |
| SILK | 20 | 0.0015 | 0.019 | 0.120 | 0.40 |
| TST [26] | 5 | 0.0011 | 0.019 | 0.115 | 0.32 |
| TST [26] | 20 | 0.0011 | 0.019 | 0.112 | 0.32 |

are prone to make highly similar predictions in this case. The longer gaps (thirty and forty-five missing frames) are more indicative of performance differences.

We present Table 1, which includes, to the best of our knowledge, the vast majority of published work that has conducted experiments on the LaFAN1 dataset. For the sake of comparability, we omit results of papers that report diverging numbers for the SLERP and RMIB baselines [5, 29] as well as works that train under different conditions, e.g. by including foot-contact as input [19] or with increased context length during training [12]. For fair comparison, we specify the primary task each model was designed for. Models not exclusively focused on motion in-betweening (MIB) as defined in [9] may show lower performance on the LAFAN1 benchmark compared to specialized models. This includes models designed for real-time applications (RT-MIB), those incorporating semantics (S-MIB), or those addressing multiple motion modeling tasks (MIB+). An examination of the related work reveals two notable observations. First, several studies have deviated from using the established set of standard missing frames (5, 15, 30, and 45) and metrics. Second, the numerical results show limited progress in the field since [26] and [24], despite many works focusing specifically on the motion in-betweening task. This lack of advancement may be attributed to a tendency among in-betweening papers to omit important state-of-the-art methods in their comparison.

## 4.4. Impact of training data sampling

As described in Section 4.1, we opt to subsample the data with an offset of 5 instead of 20 frames. While this creates a larger dataset, it also increases similarity between data points. For example, for an offset of 5, a data point of length 40 has an overlap of 35 frames with its immediate neighbor but only 20 frames for an offset of 20. For simpler models such as SILK, this additional information can guide the learning process and results in better performance as shown in Table 2. To our surprise, this is not the case for more complex models. We trained a TST [26] model using an offset of 5 during training and the publically available code. As shown in Table 2, the TST with an offset of 20 performs as well or better than the TST with an offset of 5. We hypothesize that TST, with its larger capacity, overfits slightly to the training data when sampling with an offset of 5. It might therefore be of importance to adjust the training data construction based on the model's capacity.

## 4.5. Impact of Feature Design Choices

The comparable performance of SILK, relative to more complex models, is primarily due to the larger quantity of training data, improved frame representation, and specific design choices. This section explores three key aspects: pose representation (Section 4.5.1), the choice of handling missing frames (Section 4.5.2) and the impact of using velocity features (Section 4.5.3).

### 4.5.1. Impact of Pose Representation

In this section, we explore the importance of feature representation in pose data. Unlike traditional in-betweening methods that utilize local-to-parent space features, we train SILK using root space features. To evaluate the effect of feature representation, we also trained SILK with local-to-parent space features as both input and output. The results, shown in Table 3, demonstrate that feature representation

Table 3. Comparing L2P, L2Q, and NPSS Metrics on the LaFAN1 Dataset using different pose representations.

| Length (frames) | 5 | 15 | 30 | 45 |
|-----------------|-----|-----|-----|-----|
| \multicolumn{4}{c}{L2P ↓} | | | | |
| root-space | 0.13 | 0.38 | 0.83 | 1.59 |
| local-space | 0.16 | 0.46 | 1.0 | 1.95 |
| \multicolumn{4}{c}{L2Q ↓} | | | | |
| root-space | 0.107 | 0.27 | 0.50 | 0.79 |
| local-space | 0.11 | 0.3 | 0.56 | 0.84 |
| \multicolumn{4}{c}{NPSS ↓} | | | | |
| root-space | 0.0012 | 0.018 | 0.105 | 0.30 |
| local-space | 0.0013 | 0.02 | 0.12 | 0.37 |

Table 4. Comparing L2P, L2Q, and NPSS Metrics on the LaFAN1 Dataset using different filling of missing frames.

| Length (frames) | 5 | 15 | 30 | 45 |
|---|---|---|---|---|
| L2P ↓ | | | | |
| zeros | 0.13 | 0.37 | 0.83 | 1.59 |
| slerp | 0.2 | 0.41 | 0.91 | 1.90 |
| L2Q ↓ | | | | |
| zeros | 0.11 | 0.27 | 0.50 | 0.79 |
| slerp | 0.14 | 0.28 | 0.55 | 0.94 |
| NPSS ↓ | | | | |
| zeros | 0.0012 | 0.018 | 0.105 | 0.30 |
| slerp | 0.002 | 0.019 | 0.12 | 0.35 |

Table 5. Comparing L2P, L2Q, and NPSS Metrics on the LaFAN1 Dataset using different velocity configuration.

| Length (frames) | 5 | 15 | 30 | 45 |
|---|---|---|---|---|
| L2P ↓ | | | | |
| Input Vel | 0.13 | 0.38 | 0.83 | 1.59 |
| Full Vel | 0.14 | 0.39 | 0.83 | 1.82 |
| Static Only | 0.16 | 0.51 | 1.0 | 2.1 |
| L2Q ↓ | | | | |
| Input Vel | 0.11 | 0.27 | 0.50 | 0.79 |
| Full Vel | 0.11 | 0.28 | 0.51 | 0.80 |
| Static Only | 0.12 | 0.34 | 0.6 | 0.92 |
| NPSS ↓ | | | | |
| Input Vel | 0.0012 | 0.018 | 0.10 | 0.30 |
| Full Vel | 0.0013 | 0.019 | 0.11 | 0.37 |
| Static Only | 0.0016 | 0.022 | 0.13 | 0.4 |

significantly impacts model performance, with errors increasing notably when using SILK local-space compared to the SILK trained on the features in the root space.

### 4.5.2. Impact of Filling Missing Frames

We explore the implications of two approaches for handling missing frames: replacing them with linear interpolation values as the vast majority of related work and filling them with zeros. Linear interpolation involves computing values between the last context keyframe and the target keyframe for both positions and rotations, utilizing spherical linear interpolation (SLERP) for rotations. The results, as shown in Table 4, demonstrate that SILK with zero-filled middle frames outperforms the SILK filled with SLERP for middle frames across all lengths, with the performance gap widening as the length increases. This might be caused by the SLERP input biasing the model to predict values close to the interpolated values.

### 4.5.3. Impact of Using Velocity Features

A small number of in-betweening works have used velocity features as input to their models. These include both joint velocity [29, 32] and root velocity [15] features. While modeling velocity has proven highly beneficial for motion prediction [21], its application in in-betweening has been considerably less explored.

In our experimental setup, we investigated the impact of velocity features on motion prediction performance. While position and rotation features capture the static pose information at each frame, velocity features can provide valuable information about the dynamic aspects of motion. We calculate these velocity features (linear velocity for positions and angular velocity for rotations) using two consecutive frames - specifically, requiring one additional frame after both the start and target keyframes. This velocity calculation approach means that during inference, animators need to specify one additional frame after their keyframes

to enable velocity computation. While this creates an extra requirement for the animation pipeline, we hypothesized that the temporal information provided by velocity features could significantly improve motion prediction quality.

To validate this hypothesis, we conducted an ablation study with three configurations:

1. Static features only: using joint positions, joint rotations, and global trajectory information for both input and output [Static Only]
2. Mixed features: using static features plus velocities as input, while predicting only static features as output [Input Vel]
3. Full features: using both static features and velocities for input and output, requiring the model to predict all features including velocities [Full Vel]

Table 5 presents the quantitative results for these configurations. The results demonstrate that incorporating velocity features as input while not explicitly predicting them (Input Vel) yields the best performance across all metrics. This improvement can be attributed to the velocity features providing valuable temporal information about the motion dynamics, helping the model better understand the movement patterns and transitions. However, explicitly predicting velocity (Full Vel appears to make the learning task more challenging without providing additional benefits, potentially due to the increased complexity of the output space and the accumulation of prediction errors in velocity estimates.

### 4.6. Additional Temporal Signals

While our model uses relative positional encoding for handling variable-length sequences, we investigated whether additional temporal signals could enhance the model's un-

derstanding of sequence structure. Specifically, we were inspired by the key-position embeddings from [26], which provide explicit information about proximity to key frames. For our project setup of setting middle frames to zero, this additional signal did not make a noticeable improvement for lengths seen during training. However, for extrapolating or processing sequences longer than those seen during training, it significantly degraded performance.

This result suggests that the combination of relative positional encoding and our input structure already provides sufficient temporal information. During training, we set the middle frames (which need to be predicted) to zero while keeping the boundary frames at their actual values. This creates a clear pattern where the model can identify:

- Known boundary frames through their non-zero values
- Frames to be predicted through their zero values
- Relative positioning through the positional encoding

The model appears to learn temporal relationships effectively from this implicit structure without requiring explicit signals about proximity to key frames. This finding indicates that when using relative positional encoding, the natural contrast between zero and non-zero values in the input sequence provides sufficient context for temporal understanding and generalization.

To further examine this hypothesis, we tested adding key-position embeddings when the middle is filled with slerp values. In this case, this additional signal seems to be helpful to the model and improved performance slightly for some lengths.

Table Table 6 shows a comparison of results with and without the use of key-positional embeddings as an additional temporal signal when filling missing frames with zeros and SLERP values, respectively.

## 5. Future Work

Currently, SILK does not capture the probabilistic nature of human movements. Since multiple realistic solutions can exist for any given context and target poses, a model capable of generating several solutions is preferred. Recent advances in diffusion-based models [3, 34] offer promising directions for modeling the probability density of human movements. Another area for future research is evaluating models across multiple datasets. While many studies use various datasets like Human3.6M [13], Anidance [31], and the Quadruple motion dataset [39], there is a need for a dataset specifically designed for in-betweening. Finally, developing standardized metrics to assess motion data quality and its impact on model performance is crucial. This could establish guidelines for data collection and preprocessing in motion synthesis tasks.

Table 6. Comparing L2P, L2Q, and NPSS Metrics on the LaFAN1 Dataset with and without the use of key-positional embeddings (KeyPos) when the missing frames are filled with zeros or slerp.

| Filling | KeyPos | Length (frames) | | | |
|---|---|---|---|---|---|
| | | 5 | 15 | 30 | 45 |
| | | L2P ↓ | | | |
| zeros | without | 0.13 | 0.37 | 0.83 | 1.59 |
| zeros | with | 0.13 | 0.38 | 0.82 | 2.28 |
| slerp | without | 0.22 | 0.41 | 0.91 | 1.91 |
| slerp | with | 0.17 | 0.39 | 0.87 | 1.92 |
| | | L2Q ↓ | | | |
| zeros | without | 0.11 | 0.27 | 0.50 | 0.79 |
| zeros | with | 0.11 | 0.27 | 0.50 | 0.93 |
| slerp | without | 0.14 | 0.28 | 0.55 | 0.95 |
| slerp | with | 0.12 | 0.28 | 0.53 | 0.90 |
| | | NPSS ↓ | | | |
| zeros | without | 0.0012 | 0.018 | 0.105 | 0.30 |
| zeros | with | 0.0012 | 0.018 | 0.104 | 0.35 |
| slerp | without | 0.0020 | 0.019 | 0.120 | 0.35 |
| slerp | with | 0.0015 | 0.019 | 0.110 | 0.33 |

## 6. Conclusions

In this work, we proposed a simple Transformer-based model for animation in-betweening. We demonstrated the effectiveness of our in-betweener in generating intermediate frames that seamlessly interpolate between existing key-frames. Compared to other state-of-the-art models, our approach achieves comparable or better results while employing a simpler and more efficient model architecture and without requiring elaborate training procedures.

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
