# OpenReview forum: "SILK: Smooth InterpoLation frameworK for motion in-betweening"
_thecvf.com/CVPR/2025/Workshop/HuMoGen — CVPR 2025 Workshop HuMoGen Submission_

### Official Review · Reviewer_ymod · 2025-03-26
**SILK: Smooth InterpoLation frameworK for motion in-betweening**

**Rating:** 4
**Confidence:** 4

**Review:**

Summary:
This paper presents an extensive study on the influence of neural network structure, feature engineering, and data processing on the motion in-betweening task. Specifically, the authors demonstrate that a simple transformer architecture, when combined with specialized feature and data engineering, can achieve motion transition results comparable to or even better than previous methods. While some of the claims reflect practical tricks that are already well known in the graphics community, highlighting these experimental insights is still valuable. Such findings can help researchers and engineers in this field prioritize key aspects that genuinely enhance motion quality.

Clarity and References:
The paper is well-written and easy to follow. The references appear appropriate and relevant. However, I suggest the authors include one additional related work:
"Towards Robust Direction Invariance in Character Animation" (PG 2019), which also investigates the impact of pose representations on motion generation. Including this reference would strengthen the paper’s contextual foundation.

Technical Correctness:
The technical descriptions in the paper appear mostly accurate. However, I recommend revising the claim in line 237:
“... although IK computation ...”
Based on my knowledge, local joint transformations can be computed efficiently and do not significantly increase the computational budget. Clarifying this point would make the claim more precise.

Validation:
The experimental validation is generally solid. However, I suggest training a smaller TST model on the Lafan1 dataset to evaluate whether using smaller slicing offsets could improve the model’s performance. This additional experiment could offer further insights into the model’s sensitivity to data slicing strategies.

Originality and Significance:
In my view, many of the findings in this paper reflect practical engineering knowledge that is familiar to experienced animation researchers. It is widely acknowledged in the field that a relatively simple neural network, when combined with well-designed engineering practices, can outperform more complex models with poorly executed implementations. Pioneering works such as PFNN and MANN have already demonstrated that impressive results can be achieved without overly sophisticated architectures. Additionally, some of the findings, such as using root-space features and incorporating velocity, have already been applied in earlier works from the graphics community.

That said, publicly documenting these insights is still valuable, as it can help more practitioners adopt effective implementation techniques and direct their efforts toward genuinely impactful technical improvements.

Considering the solid experimental results and the practical relevance of the findings, I recommend a weak accept for this submission.

---

### Official Review · Reviewer_aHhP · 2025-03-27
**Limited significance and novelty of the task and method**

**Rating:** 2
**Confidence:** 4

**Review:**

This paper introduces SILK, a simplified Transformer-based approach for motion in-betweening. Unlike previous transformer-based works that use complex architectures, SILK employs a single Transformer encoder to achieve comparable or superior results while reducing architectural complexity.  However, the paper is not offering anything novel in the scope of algorithms or approaches other than simplifications and design choices evaluation of existing methods.  Some evaluation results are not so convincing, especially ones about the increased trainset size upon which the performance of SILK is largely improved.  Moreover, the scope of the task is relatively limited regarding the capabilities of concurrent motion synthesis models. As a result, I would recommend weakly reject to this paper.

---
**Quality**:  Fair.

**Clarity**:  Fair.

**Originality**: Lack of originality.

**Significance**: Not so significant.

---

**Pros**:

- This works proposes a simplification of existing approaches using only a single Transformer encoder and carefully designed motion representation, showing improved performance on LaFAN1 dataset with comparison to previous works.
- Intensive ablation studies are carried out on evaluating the impact of several design choices of the model setting, including pose representation, features, and training data sampling.

**Cons**:

- The task itself - “the deterministic motion in-betweening” as stated by the author in line 129 -  is somehow out-of-date and limited, as it seems to be constrained to completing a motion sequence of only a single action (e.g. walk, run, catch a ball). And the demonstrated cases in this paper are of at most 45 frames to be filled / no more than 60 frames for the completed sequence, which are of very short sequences. At the same time, however, concurrent human motion generative models are capable of more flexible and fine-grained motion in-betweening generation with diverse poses.
- The paper is not offering anything novel in the scope of algorithms or approaches other than simplifications and design choices evaluation of existing methods.
- As shown in Table 2 and Section 4.4, it seems that the SILK method benefits a lot from the dense data sampling of LaFAN1 dataset. With the same (smaller) size trainset as it’s originally used in TST, the SILK performance is inferior, while with 4 times larger trainset, the SILK performance is superior. However, more settings of the sampling offsets should be tested and evaluated, as the performance curves of TST and SILK w.r.t the size of the trainset would be more convincing.
- In Table 1, it is unclear how the author guarantee a fair comparison. For example, is the data sampling offset consistent for all the methods?
- Lack of visual demonstrations. The shown visualization demos in the supplementary are limited. Also, the visual comparisons of TST and SILK mentioned in line 373-376 are not found in the supplementary videos.
- Some presentation issues:
    - line 326 “two different datasets” while only LaFAN1 is mentioned;
    - No indicators for the best results in Table 2-6, which makes it hard to follow.
    - line 373-376: the visual comparisons of TST and SILK mentioned are not in the supplementary videos.

---

### Meta-Review · Area_Chair_RDwW · 2025-03-31

**Recommendation:** Accept

**Metareview:**

This paper introduces SILK, a simplified Transformer-based approach for motion in-betweening that achieves strong results with a single Transformer encoder.

Strengths
----
- Shows that a simpler architecture with careful feature engineering outperforms more complex approaches on LaFAN1
- Thorough ablation studies on pose representation, features, and data sampling strategies
- Documents practical engineering insights valuable to the community

Weaknesses
----
- Performance improvements partially dependent on increased training data
- Visual comparisons could be more comprehensive
- Some presentation details need clarification

Recommendation
-----
The paper makes a valuable contribution by systematically analyzing what truly matters in motion in-betweening implementations. While not introducing novel algorithms, it provides important engineering insights that can guide researchers toward more efficient solutions.
The experimental validation is solid, and the findings will likely influence future work in the field.
The paper received mixed reviews, with one weak accept and one weak reject. I recommend acceptance, with the suggestion that authors clarify comparison protocols and enhance visual demonstrations in the final version.

---

### Decision · Program_Chairs · 2025-03-31

Accept